# A Transdisciplinary Approach to Water Access: An Exploratory Case Study in Indigenous Communities in Chiapas, Mexico

**Janeet Rondón-Sulbarán** [1,*] , **Ian Balam** [2] **and Michael Brennan** [1]

1 Department of Management, Leadership and Marketing, Business School, Ulster University, Shore Road, Newtownabbey, Co., Antrim BT37 0QB, UK; m.brennan@ulster.ac.uk
2 Fundación Cántaro Azul, Calle Franz Bloom, No. 4, Barrio de Cuxtitali, San Cristóbal de las Casas C.P. 29230, Mexico; ian@cantaroazul.org
* Correspondence: j.rondon-sulbaran@ulster.ac.uk; Tel.: +44-(28)-90368729

**Abstract:** In this article, we address the water challenges faced by Indigenous communities (ICs) in the context of Sustainable Development Goal 6 (SDG 6). The importance of ICs for sustainable regional development is increasingly recognised both at, a policy level and in practice. However, there is a reported lack of empirical evidence that gives voice to how such communities perceive development, particularly sustainable water development as articulated in the United Nations Sustainable Development Goals (2015–2030). This article provides case-based evidence from one Indigenous community setting, which challenges assumptions concerning water-related development. Using a qualitative methodology, this case study explores the problem of access to safe water faced by ICs and applies a value cocreation framework based on service-dominant logic. The findings highlight the need to develop sustainable water service models focused on the provision of water not just as a commodity or resource but also as a service. Initiatives aimed at addressing water-related challenges will be more likely to succeed when the culture, experiences, knowledge and practices of the communities in need of clean water access are valued and meaningfully incorporated into value cocreation processes.

**Keywords:** Indigenous communities; water access; transdisciplinary; value cocreation; sustainable development





## 1. Introduction

The 17 Sustainable Development Goals (SDGs) adopted by the United Nations (UN) member states in 2015 set out a framework for the next 15 years for the elimination of poverty, a reduction in climate change, and the creation of an inclusive and sustainable society for all [1]. In 2010, the UN General Assembly recognised the human right to water [2]. SDG 6 is the dedicated water goal proposed within the SDG guiding policies, relating to access to clean water for all around the world. However, there are concerns about vulnerable populations who are at a higher risk than others of "being 'left behind' in terms of water" [3] (p. 31), including Indigenous populations in Latin America. Estimates indicate that there are approximately 370 million Indigenous Peoples across 70 countries worldwide and that they account for around 5% of the global population [4]. According to the UN, in terms of most indicators of wellbeing, including access to water supply and sanitation services, ICs are usually at a disadvantage as compared with non-Indigenous populations [3]. In Latin America and the Caribbean, only 65% of the population has access to safely managed water supplies [5], compared with 99% basic water coverage in Northern America [6]. The situation is exacerbated in rural areas where eight out of ten people live without access to a basic service [6]. In Mexico, access to piped water on premises differs between 97.2% in urban areas and 85% in rural areas [7], and those most affected are Indigenous rural communities where 59.7% of the population lacks access to these basic services [8].

Thus, in Latin America, the provision of universal access to water and sanitation under SDG 6 is proving challenging. Generally, water scarcity is driven by several environmental and socioeconomic factors [9], as well as governance and water management policies that may lead to fragmentation of the water supply and distribution services [3]. One way to narrow the gap in water access is through the development of low-cost technologies. However, technological innovation does not guarantee a solution to complex real-life challenges. There is abundant literature that has discussed the factors impacting the limited success of these innovations. Evidence suggests that the failure of effective systems and infrastructure installed to provide safe, quality water to households in rural communities in Latin America is attributed to a persistent use of unsafe water sources by the beneficiaries of such systems [10,11]. Additionally, Brown and Pena [12] found that the lack of success of technological innovations is impacted by a limited understanding of the motivations for their adoption or a lack of knowledge about the impact of these interventions in rural communities. In the same vein, Thurber et al. [13] reported on how the sole assumption of health impact is not a sufficient motivation for the use and adoption of these technologies. The authors argue that redesigning the technologies according to an understanding of what users value beyond health would make them more likely to succeed [13].

Transdisciplinary research (TDR) has emerged as one approach to understanding and addressing such 'real-world' challenges. This approach encourages the coproduction of knowledge and engages scientific and societal partners in a collaboration that challenges assumptions and privileges the demonstration of actual impact [14,15]. TDR is increasingly being adopted in sustainable development research and, in particular, water-related challenges [16–18]. Furthermore, in recent decades, changes in the development of business and marketing theories have placed value cocreation as an alternative to creating sustainability in the water sector [19]. The dominant logic in marketing has evolved from the exchange of tangible goods to a service-dominant logic engaging with the cocreation of value [20]. In this study, we would argue that typologies of value cocreation practices could be a useful way to better understand ICs in the context of addressing water-related challenges. In Mexico, for example, where 85% of rural areas have access to piped water, diarrhoeal disease is still the highest risk of death for children under five years of age [21]. Recently, the World Health Organization (WHO) estimated that 60% of all deaths caused by diarrhoea in low- and middle-income countries are attributable to unsafe drinking water, sanitation and hygiene [22]. Despite these facts, there has been scant research on both the water-related challenges faced by rural communities such as those presented in this study and what solutions best serve their needs.

In this paper, we describe the perceived water-related problems in three ICs in Chiapas, Mexico and how to address those issues. In these contexts, different funding bodies and organisations offer development interventions or low-cost technologies as viable solutions to the problem of access to safe water. These initiatives are usually implemented through local non-governmental organisations (NGOs) with an established presence in the communities. However, while anecdotal evidence suggests positive outcomes, these organisations and sponsors also perceive a lack of changes in diarrhoeal disease, particularly in children. To understand the implications of low-cost technologies for safe water, we present the case study of three communities in Los Altos, Chiapas. Data were collected through focus groups with representatives from the three communities as well as other stakeholders, including government agencies, local authorities and NGOs engaged with these communities. To understand how the needs of the communities can be translated into practice, we relate the issues that emerged from the focus groups to existing frameworks of value cocreation practice in order to present a model of water as a service for ICs in Chiapas. The paper is structured as follows: after the Introduction, we describe the conceptual approach used and briefly relate this to water-related challenges impacting Indigenous Peoples. Then, we describe the case study setting and explain the methodology. This is followed by the findings in which the key issues impacting safe water access are presented,

with potential solutions based on a customer value cocreation practice styles framework outlined in the discussion. Conclusions are presented in the final section.

## 2. Conceptual Approach

*Service-Dominant (S-D) Logic and Water-Related Challenges*

In recent decades, 'service logic' has evolved from the traditional goods or product logic to the cocreation of value across a wide range of practice-based disciplines. Value cocreation is emerging as an important condition for actually sustaining interventions aimed at addressing local water challenges [19]. In this sense, Pahl-Wostl et al. commented on the increasing number of water challenges reported over a decade of water research. The authors suggested that the solutions to these problems should be based on interventions focused on the coproduction of knowledge at different levels and scales, involving both scientists and societal stakeholders [23].

Further, assuming that water has a value beyond its modern utilitarian perception, we would argue that ICs are not passive recipients of water services. With the emergence of S-D logic, Vargo and Lusch [24,25] present a new understanding of customers or beneficiaries as active participants in the value and service chain. S-D logic refers to a new paradigm in marketing thinking, which establishes that "marketing has moved from a goods-dominant view, in which tangible output and discrete transactions [are] central, to a service-dominant view, in which intangibility, exchange processes, and relationships are central" [24] (p. 2). This theory is based on a series of foundational premises. One of these premises (FP6) indicates that the customer is always a cocreator. In other words, "the consumer is always involved in the production of value" [24] (p. 11). This implies, as Vargo and Lusch have indicated, that "*value is cocreated*, rather than created by one actor and subsequently delivered" [20] (p. 47). Under this principle, the customer is considered mainly a coproducer rather than a 'target' and can be involved actively in the whole value and service chain [25].

Likewise, scholars have recognised the interactive and iterative nature of value cocreation [26,27] and the transdisciplinary nature of S-D logic [20]. Theoretical studies have highlighted the dynamic and contextual nature of value cocreation and the role of resource integration, which have led to empirical research across diverse service contexts [26,28,29]. Resource integration focuses on ways in which individuals engage with others in their service network to integrate resources [29]. Given this novel reconfiguration of products and services, and the key role of resource integration, S-D logic can potentially be applied in the context of water challenges in ICs to encourage solutions that could fulfil different needs or wants through a process of value cocreation. Water is the common denominator that links nearly every SDG. Moreover, ICs have particular worldviews that afford water a prominent place in their life and culture [30–32]. Such communities possess traditional knowledge and distinctive ways of being and believing that give them a spiritual connection to nature and all forms of life [30,31,33]. Their water-related knowledge and traditional responses to challenges are now more important than ever [34].

In agreement with McColl-Kennedy et al.'s interpretations of S-D logic, in this paper, we have adopted their definition of value cocreation as "benefit realized from the integration of resources through activities and interactions with collaborators in the customer's service network" [29] (p. 370). Drawing on existing S-D logic theory [24], McColl-Kennedy et al. [29] developed a framework that distinguished practice styles in value cocreation to improve quality of life, which was applied in a healthcare setting. The authors divided the construct into eight activities and correlated interactions, and established five different types of customer value cocreation practices in terms of activities and interactions actually undertaken by patients both within and outside their service network. By crossing the dimensions of activities and interactions, McColl-Kennedy et al. developed a 2 × 2 matrix. As shown in Figure 1, the typology is based on five different perceptions of the customer's role regarding two aspects: (1) low to high level of activities and (2) low to high number of interactions with different individuals in the service network [29].

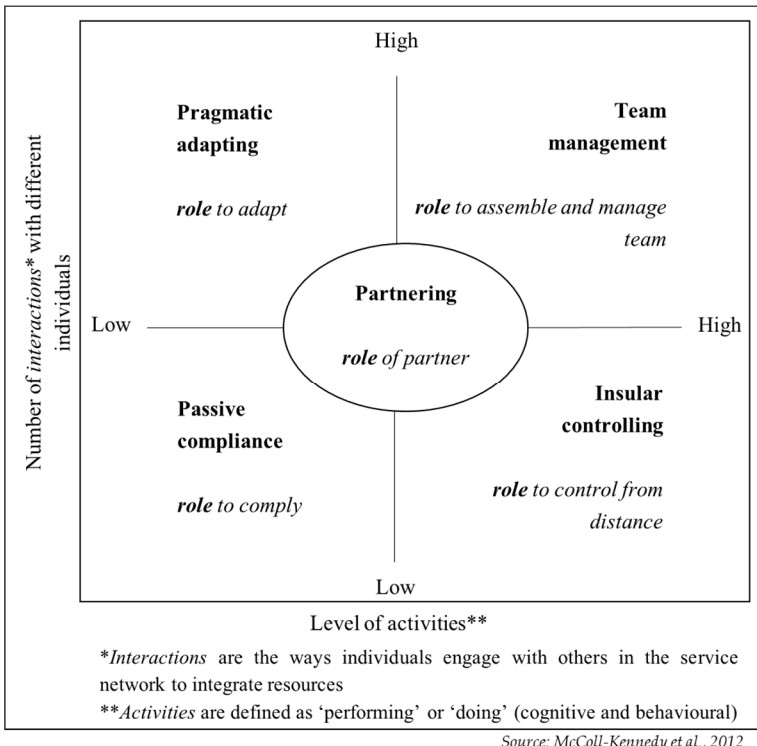

**Figure 1.** Customer value cocreation practice styles (CVCPS) framework.

In this study, we used McColl-Kennedy et al.'s [29] customer value cocreation practice styles (CVCPS) framework, in an attempt to ascertain how ICs cocreate value in the water service setting. Our research, thus, considers that different stakeholders perceive water-related challenges in different ways and that one solution does not fit all. This study, through the lens of S-D logic, tries to provide novel solutions to water challenges aligned with alternative forms of sustainable development as established by the SDG 6 for water. The next section describes the research setting and outlines the methodology.

## 3. Research Setting and Methods

Mexico is a federation of 32 states with a population of approximately 129 million people concentrated in urban areas [4]. Despite the enormous economic growth and prosperity experienced in the country in recent decades, large portions of the population still live in poverty (46.2%), with the southern and southeastern areas being particularly affected by poverty and deprivation [35]. The Mexican government has strived to resolve the precariousness caused by issues of water and sanitation and to provide universal coverage of those services. In 2012, the recognition of the citizens' human rights to water and sanitation was incorporated into law through constitutional reform [36]. The National Water Commission, known as CONAGUA by its initials in Spanish, is the decentralised agency of the Ministry of the Environment and Natural Resources tasked with the mission of achieving sustainable use of the water resource and ensuring that Mexico as a nation has sufficient water in terms of both quantity and quality [37]. In rural areas, water supply is the responsibility of water boards known as *Juntas*. Through different mechanisms enabled by government legislation, the water boards share the responsibilities of water supply management with water committees. These are management divisions charged with managing sustainable development and community organisation initiatives [38]. Government programmes such as *Prossapys* (by its initials in Spanish) (Prossapys: Programa para la Sostenibilidad de los Servicios de Agua Potable y Saneamiento en Comunidades Rurales/Programme for the Construction and Rehabilitation of Drinking Water and Sanitation Systems in Rural Areas), with the aim of increasing water coverage in rural areas through the building of infrastructure, are overseen by CONAGUA [39]. The State of Chiapas, in the southern

region (Figure 2), with over five million inhabitants, is the poorest state of Mexico and depends on exports of a few agricultural products, with no economic diversification [40–42]. Chiapas is one of the Mexican states with the largest water resources, contributing 40% of the country's total, [43]; however, approximately 70% of its population do not have access to drinking water and sanitation, and only 26% of households are provided with piped water [44]. The region has one of the highest Indigenous populations in the country, and it is the most affected in terms of provision of basic household services, including potable water [45]. Furthermore, water sources are polluted as most of the wastewater remains untreated, and 30.6% of the population are at risk of gastrointestinal disease [44].

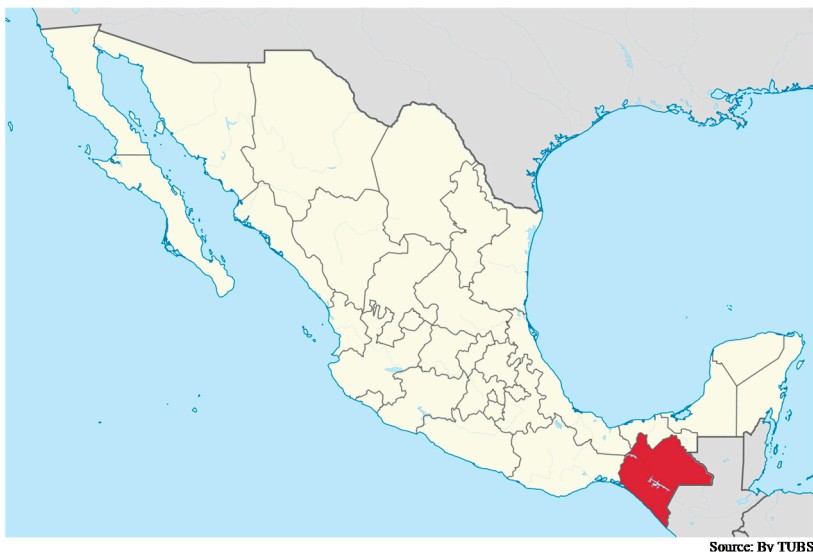

Source: By TUBS

**Figure 2.** Map of the State of Chiapas, Mexico.

In Chiapas, 27.94% of Indigenous people over 3 years old speak an Indigenous language. The most representative Indigenous groups in this region are of Maya ancestry and include the Tzeltal, Tzotzil, Ch'ol, Zoque and Tojol'ab'al [40,41]. The three Los Altos communities that are part of this study belong to the Tzeltal and Tzotzil groups, and they have been identified by the Chiapas Ministry of Health as being among the 30 poorest municipalities in the region, with high rates of the population falling into the lowest national wealth quintile [46]. The main water sources in the area are streams accessed by communities through water committees or *patronatos del agua* as part of the community-based management and supply system. Generally, the *patronatos* are controlled by males, although recently some of them are starting to accept women among their ranks [47]. Women are mainly responsible for the collection and management of water for domestic use [43].

*3.1. Data Collection*

Original data for this study were collected in 2018 during a one-week training workshop held in Mexico as a key activity of a broader TDR programme between Ulster University (UU) in the UK, the University of São Paulo (USP) in Brazil, the University of Medellín (UdeM) in Colombia, and two NGOs: Fundación Cántaro Azul (FCA) in Mexico and the Centre for Science and Technology of Antioquia/Centro de Ciencia y Tecnología de Antioquia (CTA) in Colombia. One of the priorities of this TDR programme is to enhance the social impact of water research through a meaningful involvement of stakeholders and end-users throughout the project lifespan. UU, together with the research partners, has designed a research strategy including local organisations, underserved end-users, and public and private agencies who are not only potential recipients of the knowledge generated by the programme but also participate actively in the coordination of project activities and the dissemination of findings. Since 2017, academics and researchers from

different disciplinary backgrounds, non-academic partners, stakeholders and local communities have worked on projects aimed at the development of low-cost technologies for safe drinking water, capacity building, the development of new networks and strengthening of existing ones, and the facilitation of cocreation processes, alongside local actors, who are adapted to the local realities.

Working in partnership with FCA (based in Chiapas, Mexico), the TDR programme has engaged rural communities in the development of low-cost water technologies. FCA has a track record in the implementation of water-related projects in harmony with international development practices. Using a wide range of participatory approaches, FCA involves community members in the management of local water treatment and water supply systems through tailor-made solutions consisting of water kiosks and low-cost household systems (e.g., *La Mesita Azul*/The Little Blue Table).

The three communities that participated in this study are located in the region of Los Altos where FCA operates (Figure 3). Most of the Indigenous population in Chiapas is concentrated in the Los Altos, Norte and Selva regions [48].

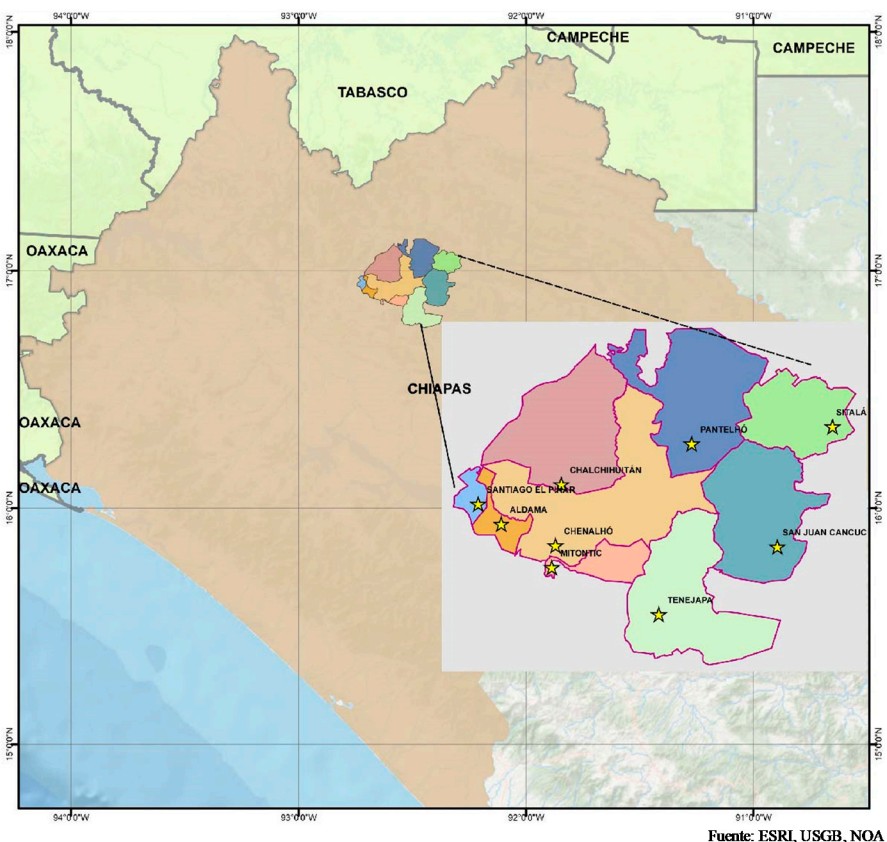

**Figure 3.** Map of Los Altos Municipality, State of Chiapas, Mexico.

The case study method adopted in this research was influenced by two factors. First, this method allows for an in-depth exploration of real-life situations [49], and second, it is suitable to answer questions related to the 'how' or 'why' that aim to identify key variables and associations. Consistent with this method, we have applied an established theoretical framework [50] drawing on S-D logic theory [29] to demonstrate how suitable value cocreation models can contribute to solutions to water-related challenges and sustainability. A series of qualitative methods were used to collect data, including focus groups, note taking during workshop activities, unobtrusive sources (e.g., websites, documents and presentations) and observations by the lead author of this paper.

The Logical Framework Approach (LFA), a valid methodology for the evaluation of complex projects [51], was used to gather data in focus groups with ICs and key stakeholders. Specifically, the method of the 'problem tree' suggested by the LFA was favoured because it offers participants the freedom of using written or pictorial language. This method, applied in the focus groups with representatives from the ICs, as well as the use of interpreters was thought to be adequate in order to minimise the language barrier—participants spoke Indigenous languages (Tzotzil or Tzeltal) and the researchers Spanish and/or English. The focus groups held with ICs and other stakeholders were not recorded; researchers took comprehensive notes during these activities. Contents of all data from focus groups (with ICs and key stakeholders) were thoroughly read and grouped into evolving thematic categories that informed further work. A triangulation method of analysis was used to compare the information gathered from the different data sources [52].

In keeping with Indigenous methodologies, issues of consent were managed, as recommended by Ellis and Early [53] following the particular culture and traditions of the participants from the ICs. This study was considered not to meet the definition of 'research' according to the national Research Ethics Committees (REC) algorithm; therefore, it was exempted from a submission for ethical approval (the SAFEWATER project has been granted ethical approval by the Research Ethics Committee (REC) in Northern Ireland, United Kingdom (case number REC/18/0064), as well as site-specific ethics committees at Fundación Cántaro Azul (FCA) in Mexico and at the University of Medellín in Colombia). During the focus groups with ICs, written consent was evidently not a high priority for the participants; however, they opted for anonymity given that they were representing a collective (their own communities) and not their individual voices. Before commencing the activity, nonetheless, informed consent was obtained verbally by entering into a reciprocal partnership [53] in which reciprocity was based on the understanding that both researchers and participants were engaging in a collaborative learning experience that could be of benefit for both. After the researchers reminded participants of the overall aim of the larger project and of the objectives of the focus group, everyone agreed to participate. Before commencing the activity, all present had an opportunity to briefly introduce themselves while sharing some snacks and refreshments brought by the researchers as a counterpart to the lunch offered to them by the participants once the interview was completed. This gesture was valued by the participants and served not only as an ice breaker, but also created an atmosphere of trust and respect within the group, also important in the conduct of research with Indigenous people [54].

### 3.2. Focus Groups

### 3.2.1. Indigenous Communities (ICs)

FCA selected a purposive sample of three ICs from Los Altos, Chiapas to participate in a focus group discussion that lasted approximately four hours. This sampling method not only helped in the selection of participants with the specific characteristics that could provide useful information [55], but also helped researchers in reducing the cultural and language barriers because the participants welcomed the intervention of FCA staff as interpreters. Also, the participants were put at ease when they met the researchers and found out that they were also natives of Latin America (Venezuela and Colombia). The 10 participants in the focus group consisted of a diverse group of representatives in terms of age, gender, roles and knowledge about the communities. All the participants were health promoters who had qualified through their participation in training programmes offered by FCA through its various water-related initiatives. One 52-year-old male participant was a member of the water board (*patronato de agua*) in his community, and the female representative held a key role in the water, sanitation and hygiene (WASH) programme in hers. The sample is described by age and gender in Table 1.

**Table 1.** Participants from the ICs by age and gender.

|  | Aged 18–25 Years (*n* = 5) | | Aged 26–35 Years (*n* = 2) | | Aged 36–45 Years (*n* = 2) | | Aged 46–55 Years (*n* = 1) | |
|---|---|---|---|---|---|---|---|---|
|  | **Male** | **Female** | **Male** | **Female** | **Male** | **Female** | **Male** | **Female** |
| Group 1 | 2 | | 1 | | | | | |
| Group 2 | 1 | | | | 1 | | 1 | |
| Group 3 | 2 | | 1 | | | 1 | | |

The focus group took place in the training room of the health centre in one of the Indigenous communities; a venue that was familiar to all the participants. Using the problem tree method [51], participants answered the general questions of 'How do you perceive the water challenges in your area?' and 'Who is/are affected by these challenges? what are the effects?'.

3.2.2. Key Stakeholders

Participants purposively selected through contacts maintained by FCA were interviewed in a focus group. The participants were experts in their area that had a history of working alongside NGOs with the Indigenous communities in the implementation of social policy programmes and other initiatives, including water supply and sanitation. The focus group of approximately 90 min comprised 10 representatives from different agencies, including the public health authority (*n* =2), health and hygiene division (*n* = 1), water services (*n* = 3), water infrastructure (*n* = 2) and NGOs (*n* = 2). Other data were obtained through formal and informal conversations and unobtrusive methods.

*3.3. Lead Author's Observations*

A native speaker of Spanish born and raised in Venezuela, the lead author has also lived for 26 years in a European country. Her appearance and ethnicity made it easier for her to blend into the research setting. Her mixed cultural background and an education in social research methods allowed her to approach the research with an open-ended view. From the onset of the training workshop, the lead author realised that an accurate representation of the culture and research setting was key in the context of the study; thus, the method of participant observation as suggested by DeWalt and DeWalt [56] (p. 99) was used to gain a "holistic understanding of the phenomena under study."

**4. Findings**

This section starts with the problem analysis as perceived by the ICs and the group of key stakeholders; added to this analysis are the observations of the lead author. Then, the negative issues reported by the participants are carefully analysed and potential solutions identified based on value cocreation practice styles.

*4.1. The Problem of Availability and Access to Clean Water*

4.1.1. Indigenous Communities (ICs)

ICs attributed the lack of availability and access to clean water to a lack of 'universal coverage' derived from the denial of access to sources by private proprietors as well as through deforestation or pollution. A diagrammatical representation of the problem analysis in Figure 4 illustrates the relevant issues with cause (bottom boxes of the diagram) and effect relationships (top boxes of the diagram).

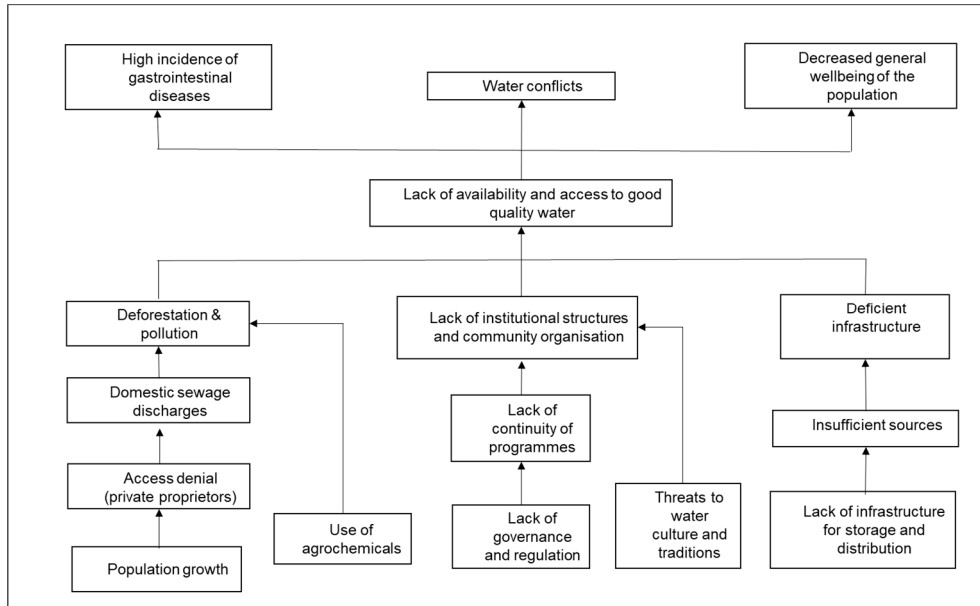

**Figure 4.** Perceptions of the problem of water by ICs in Los Altos, Chiapas, Mexico.

Water conflicts could ensue between private proprietors, resulting in communities being deprived of piped water systems, as demonstrated through this comment by one of the participants:

> The stream is in a private source and the owner denies the access to it. We can't do anything about it, he does not listen, and all he wants is the money. So, we can't have piped water like many other communities and the authorities don't do anything about it. (52-year-old male)

The above comment also hints at the resentment the communities feel towards the authorities due to the lack of governance and regulation to prevent both denial of access and pollution of the sources. Pollution is mainly caused by an increase in population and anthropogenic activities. Participants expressed great concerns about these issues and felt that they did not receive appropriate attention from the authorities, as exemplified by this quote:

> You see it all the time. People put waste in a lorry and they dump it in the streams because there hasn't been any refuse collection for weeks. (35-year-old male)

Aligned to this culture of neglect is the change of values in the communities, which seems to erode the water culture and traditions. Concerns were raised regarding intergenerational water differences that, according to some participants, also hamper community participation and organisation:

> The young people don't participate in the community and they just don't care much about water. They are not worried about not having clean water because they always carry bottled water or drink [fizzy drinks]. I just explain to people that water is better and healthier, and you can use it to make nice juices. (20-year-old male)

All these changes in habits, lifestyle and attitudes, ignoring the ancestral values and traditions of the Maya culture, are contributing to a decline in health and wellbeing and a depreciation of water in the communities, as one participant expressed:

> There is a tradition in our culture to protect our springs . . . our sources of water and we are happy when the rain comes because we know that there will be plenty of water . . . young people, they don't see that, they just go to the shop and buy a bottle of mineral water. (36-year-old male)

While the lack of standards and water regulation as well as the denial of basic services as a product of poverty in the area are partly to blame for the lack of access to quality water, participants also expressed the need to mobilise community resources and human capital in order to raise awareness and strengthen the water culture. This view was expressed by this participant, who said:

> People just drink water; they don't want to drink safe water. They don't see the danger of drinking any water. So, in order to change, you need to change people's habits first. (45-year-old female)

Participants also commented on the lack of community responsibility for the systems when community and/or household technologies for water treatment are available. While it was recognised that having these technologies gave communities a sense of cooperation, a lack of commitment among communities to manage the systems was also highlighted. Also noted was the increased waste (for example, plastic filters) these systems could generate and the difficulties in maintaining or replacing some of their parts, emphasising the unsustainability of these solutions.

### 4.1.2. Key Stakeholders

In the stakeholder group, the problem of lack of availability and access to safe water appeared to have different causes from the ones discussed by the ICs. Nonetheless, one of the main effects was the high incidence of gastrointestinal diseases, as also pointed out by the participants from the ICs (Figure 5 illustrates the perception of the problem). The key stakeholders discussed extensively the need for a change in paradigm and approaches to allow an honest analysis of the water-related challenges in the pursuit of effective solutions.

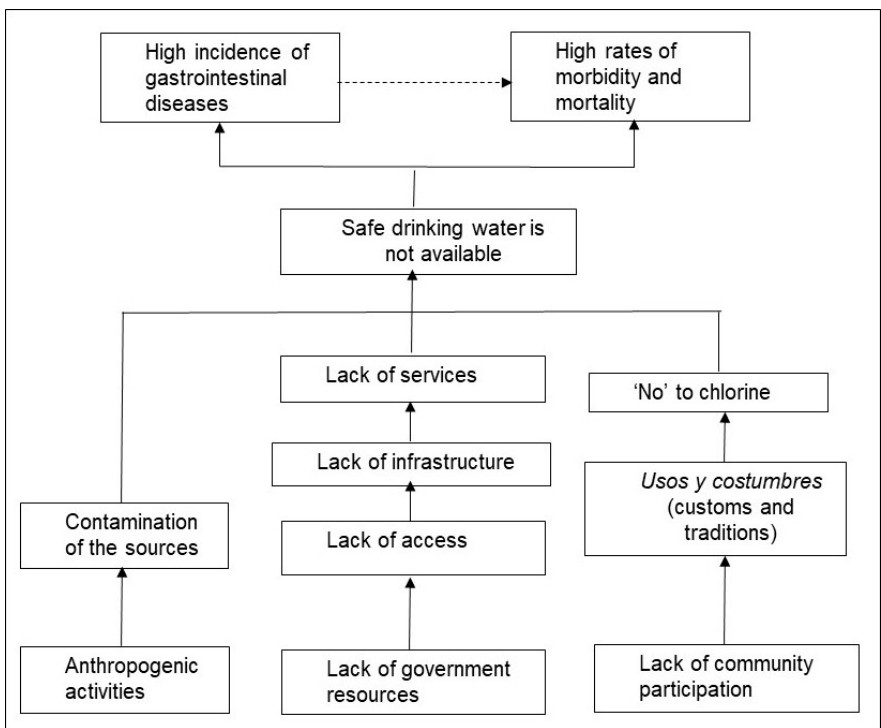

**Figure 5.** The water problem and perceptions of stakeholders working with ICs in Los Altos, Chiapas, Mexico.

The NGOs highlighted the importance of bottom-up approaches and participation and the key role of context sensitivity for resolving water challenges. Historically, approaches have tended to be top-down in practice. For example, while most participants agreed with the provision of infrastructure, they also criticised the lack of strategic planning in this provision highlighting how most projects have failed due to lack of continuity and

capacity building. Similarly, methods of disinfection, such as chlorination, were mentioned as another hurdle that prevented the adoption of many technologies. As indicated by many in the group:

> No-one in the communities would adopt such systems, while they leave a taste of chlorine in the water. (Health Authority)

Participants from government agencies suggested that uses and customs inherent to the Maya culture have contributed to the failure of chlorine disinfection systems, arguing that an understanding of these cultural mores and the genuine involvement of the communities in the design and implementation of interventions could be the pathway to success. NGOs proposed a paradigm shift towards the Maya concept of *'lekil kuxlejal'*—*'buen vivir'* or *good living, living well* or *dignified life*. From this perspective, many of the stakeholders believed that interventions that are based on the pillars of cocreation, self-management and autonomy are more likely to succeed.

### 4.1.3. Lead Author's Observations

The September 2018 Training Wrokshop of the international programme began with a blessing ceremony infused with Maya elements. This ceremony, thus, was a revelation of the deep spiritual connection between water and the people in the research setting. Attendees were gathered around an altar representing the cosmovision of the Maya people, which offers a holistic understanding of the universe where all beings live in harmony with one another. This ceremony revealed the symbolism of Earth, water and food and the connections of these elements for human survival as depicted in the *Pop Wuj* or Maya Bible. The connection with water, nature and life and the need to find a balance between these elements and people's needs was addressed by the spiritual guides conducting the ceremony. Given the sacred space where water was honoured by all present, how can low-cost technologies for safe drinking water keep the harmony and balance in these communities?

The presentations given by the NGO partner of the research programme facilitated a further understanding of the *'lekil kuxlejal'*/*'buen vivir'* concept, and the meaning of water for the Tzotzil and Tzeltal Indigenous Peoples in the region was learned. In the Maya culture, water is 'the sacred' that generates 'spirit/life.' This ancestral belief is maintained in the communities for whom the 'clear' water running down the streams or springs is safe; they drink it directly from these sources—how could the sacred element kill you?

Field visits to several communities provided further insights. The trip showed the vastness of the geographical area and the dispersion of the communities (many of less than 100 households). Families lead simple lives and maintain the traditions of their ancestors. Water is vital for their survival. Most men engage in subsistence agriculture and women look after the home and the children, engage in minor economic activities and participate in government initiatives with socioeconomic purposes, within which are that include water-related programmes. Most water infrastructure is obsolete or has decayed, and more importantly, there is no potable water. In domestic settings, basic technologies for safe water have been provided by an NGO. Mainly women, who are trained by the NGO, have responsibility for these systems, giving them a sense of empowerment.

### 4.2. Cocreating Value for Water Access, Sustainability and Development

The problem analysis helped us identify the negative aspects of the existing situation in these ICs regarding access to clean water. Thus, the major problem of lack of availability and access to safe drinking water, analysed in terms of its 'cause and effect' relationships, offered an opportunity to both ICs and key stakeholders to identify those difficulties they attach a priority to and which they wish to overcome. A number of issues, as discussed above and summarised in Table 2, were identified in both groups, and these were complemented by the lead author's observations. Taken together, these findings suggest the need to develop sustainable water service models focused on the provision of not just a product (that is, a technology for safe water) but a service, where service is defined as the "application of one's resources for the benefit of another entity" [27] (p. 28).

**Table 2.** Issues impacting availability and access to safe water from different sources.

| | Sources | | |
| --- | --- | --- | --- |
| | **Indigenous Communities** | **Key Stakeholders** | **The Lead Author's Observations** |
| Issues | • Lack of institutional structures (governance and regulation) and community organisation<br>• Access denial<br>• Threats to water culture and traditions<br>• Pollution and deforestation<br>• Lack of infrastructure and storage and distribution systems | • Top-up<br>• Access denial<br>• Cultural mores<br>• Anthropogenic activities<br>• Lack of services, infrastructure and resources<br>• Disinfection methods —'no' to chlorine | • Spiritual dimension overlooked |

The ICs are conscious of the role that water plays in their everyday life, not as a commodity, but as a natural element with multifaceted implications for the communities; therefore, its preservation appeared to be one of their main priorities. These are the elements that need to be an integral part of water models that consider water as a service and not just as a mere commodity. In such models, technology is a complementary element to the service that also adds value. The key stakeholders seem to share similar views to those of the ICs. Many in this group expressed the need to reduce the high incidence of gastrointestinal diseases. Stakeholders involved in service provision firmly believed that alternative solutions to disinfection through chlorination needed to be found as people rejected the smell and taste of chlorine left in the water. In the communities, it was recognised, as expressed by one participant, that 'a change in people's habits' was needed in order to make people understand the dangers of drinking untreated water. These are also elements that are needed to complement the proposed models, which could be incorporated through education, capacity building and awareness activities. A common view amongst participants was the need to reduce the incidence of gastrointestinal diseases as one of the most detrimental effects of the lack of safe water. A comparison of the issues prioritised by all participants (as shown in Table 2) reveals a convergence of views, which can increase the opportunities for all stakeholders to identify potential specific goals and objectives that could lead to adequate solutions to the lack of access to safe drinking water in these communities. The findings suggest the enormous potential for the development of water service models using existing frameworks for practice styles of value cocreation [29]. In other words, the different stakeholders in this study (including the ICs as a stakeholder) can potentially cocreate value with the integration of resources available to the different groups to reach the ultimate goal of achieving the provision of safe drinking water according to their needs. They may have the same value styles as the ones described by McColl-Kennedy et al. [29], or a different range of styles may emerge that would better display the characteristics of the water area setting.

The observations of the lead author identified the centrality of the different values attributed to water (e.g., spiritual dimension). Most participants from both focus groups conceded that the belief in the sacred nature of water was primordial in the communities. In this context, this insight should be given high priority in any potential water service model. A striking example of this belief is a famous tourist attraction in the region—a waterfall called the 'Christmas Tree.' It is named as such because water cascades over the vegetation in the shape of a Christmas tree adhered on the walls of a 1000 mt cliff. This is an impressive view that has fascinated all inhabitants of these ancestral lands for generations. This view coupled with the religious Maya ceremony held at the commencement of the training programme, emphasised the intimate connection of these communities with water, which has been described by Chaves López [57] as a unique perspective that . . .

[ . . . ] *considers the supernatural beings of the water sources, predominantly the ladies or mothers who inhabit them, and with whom community sostain (sic) a permanent relationship that promotes fisical (sic) as well as ritual caring for water and territory to be inherited to future generations.* (italics in the original, p. 21)

## 5. Discussion

This study set out with the aim of exploring issues related to lack of access to safe drinking water in ICs in southern Mexico. Having acquired a holistic view of the problem, we suggest testing the applicability of McColl Kennedy et al.'s [29] customer value cocreation practice styles (CVCPS) framework to the water service area (recall Figure 1). The CVCPS framework offers mechanisms through which ICs (or other local communities or vulnerable groups), academics and researchers with diverse disciplinary backgrounds, as well as other key stakeholders (e.g., local authorities, government agencies, policy makers, NGOs, funders etc.) can cocreate value in practice, through activities and interactions amongst themselves and other collaborators. The advantage of this framework is that end-users or beneficiaries (i.e., customers) play a key role in these exchanges as they actively participate in the process and decision making.

In the applicability of the framework, not all the practice styles are transferable to the water setting and, given the different contexts (e.g., social, economic, political and environmental) considering the different rural communities, it is likely that other value cocreation styles may emerge. To make safe drinking water available and accessible under the assumption that the ICs display a team management CVCPS (a high level of activities and a high number of interactions with different actors or stakeholders), it is expected that they will engage in activities and interactions that can address the issues and difficulties causing the problem. Thus, to mitigate the impact of deforestation and pollution on the water sources, the development of a water quality monitoring strategy, as well as a strategy to halt deforestation, should be considered a priority. It is possible that legal regulations are already established to prevent deforestation and pollution, but they are not enforced. This implies identifying why the institutional structures are failing to reverse the presumed non-compliance. In order to do this, the ICs could be expected to engage in activities such as collating information, learning and connecting (e.g., with local authorities, policy makers and service providers), which, in turn, involves interactions with individuals from the same community and/or other communities, or with agencies from different sectors (i.e., public, private, voluntary or academic). Similarly, the communities can be expected to engage in coproduction activities such as the participation in the design, implementation and adoption processes of any clean water system that they deem to be appropriate. In order to do this, key aspects of the lifestyle and social world of the ICs, such as cultural mores, traditions and beliefs related to the value of water, should be central to the cocreating activities.

What can be done to overcome the challenges for the successful implementation of low-cost technologies in developing regions? Certainly, in terms of the provision of low-cost safe drinking water technologies in line with the SDG 6 for water and sanitation, there is more scope for the application of frameworks based on social science or social practice theory alongside the development, deployment and implementation of technological solutions. Scientists and water specialists can help with the provision of technical information, knowledge and expertise, while the social science professionals can delve into the 'what', 'why' and 'how' of the issues at stake and complement and enhance the potential technical solution. However, any agencies or groups of stakeholders involved in water service provision in settings and contexts like the one described in our case study need to be cognizant that in this type of value cocreation process, the concerns, ideas, beliefs, knowledge and expertise of the vulnerable groups for whom the solutions are intended vary, and one size does not fit all.

## 6. Conclusions

Safe water provision in ICs constitutes an enormous challenge that has not been addressed adequately in the context of SDG 6 for water and sanitation. Based on how closely interlinked water is with sustainable development, innovative approaches to tackle this problem have become a priority. This study has addressed this issue and has used an existing framework for value cocreation practice styles that is useful in understanding how ICs can instigate innovative water service models. These value cocreation practice styles suggest that there is not a single solution for water challenges and that those interested in addressing these issues have to engage with the problem in different ways and consider the different values attached to water. The collected evidence from focus groups, observation, as well as the analysis of unobtrusive sources has demonstrated how ICs prioritise the spiritual, cultural, historical and environmental dimensions in their life. From a practical point of view, any organisation and agency delivering programmes or initiatives in these communities, as well as research funders of development initiatives, should consider carefully in the first instance, the culture, traditional knowledge and relationships built based on trust and mutual learning. Any initiative aimed at research addressing water-related challenges will be more likely to succeed when the experiences, knowledge and practices of the communities are valued and meaningfully incorporated into any solution. Having a holistic understanding of how water is viewed by these communities and its impact on their daily lives can break new ground, not only in reframing the solution to the complex problem of safe drinking water provision, but also in redefining approaches to development in ICs. Potential water service models in these contexts, focused on the provision of not just a product (that is, a technology for safe water) but a service, defined as "the application of one's resources for the benefit of another entity" [26] (p. 28) would generate enormous opportunities for societal and scientific partners to embark in meaningful and long-term partnerships, collaborations and programmes.

The qualitative approach adopted in this study and the methods used to collect data, particularly the 'problem tree' for focus groups, proved to be a very effective technique. This gave participants the opportunity to engage in discussion using written or pictorial language. Participants from the IC group valued the opportunity to engage in discussion with individuals from different communities and be able to learn about everyone's problems and how they could be solved. Some of the participants commented on how they could take away this learning and apply the technique in discussion groups in their communities. Using a TDR approach, we have uncovered some of the real-life water-related problems that ICs can face and understood them from different perspectives. The learning of the deep rootedness of Indigenous culture with nature and the environment offers scientists the opportunity to embark on a capacity building journey enriched by other forms of wisdom and knowledge beyond that provided by scientific development and modernity, acknowledging, as Acosta has said, "the call to put technology at the service of life instead of capital accumulation" [58] (p. 2607). Finally, the reader should bear in mind that the findings in this study are limited to a single study case; however, many other Indigenous Peoples in Mexico and across the region live in similar circumstances to these communities in Los Altos de Chiapas. Notwithstanding this limitation, the findings contribute to the literature of value cocreation with the applicability of a typology of value styles in the water service area.

**Author Contributions:** The contributions of the three authors named in this manuscript consist of the following: Conceptualization, M.B. and J.R.-S.; Methodology, M.B. and J.R.-S.; Investigation, J.R.-S.; Writing—original draft; J.R.-S.; Writing—review & editing, I.B. and M.B. All authors have read and agreed to the published version of the manuscript.

**Funding:** This research was funded by the Global Challenges Research Fund (GCRF) UK Research and Innovation/Engineering and Physical Sciences Research Council (EPSRC) under Grant number EP/P032427/1.

**Institutional Review Board Statement:** The SAFEWATER project has been granted ethical approval by the Research Ethics Committee (REC) in Northern Ireland, United Kingdom; case number REC/18/0064.

**Informed Consent Statement:** Informed consent was obtained verbally from all subjects involved in the study.

**Data Availability Statement:** Not applicable.

**Acknowledgments:** The authors would like to thank the consortium members and partners (www.safewater-research.com) who participated in this study.

**Conflicts of Interest:** The authors declare no conflict of interest.

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
