# Peer review of "A Transdisciplinary Approach to Water Access: An Exploratory Case Study in Indigenous Communities in Chiapas, Mexico"

_water, doi:10.3390/w13131811_

Round 1
Reviewer 1 Report
The paper makes an attempt to adress the water challenges faced by indigenous communities in Mexico. The authors adopted a qualitative approach to present their findings. The findings are well explained. I have few concerns which are given below:
- The number of participants is not mentioned any where in the paper. It is not clear how many key stakeholders and lead authors participated in the study. More details about the position of the key stakeholders and lead authors will help to understand the perceptions of the respondents.
- The authors mentioned in the conceptual framework section, "ICs cocreate values for development" (Line 147). I assumed that the identified values will help in suggesting water development in the study area. However, as the values are identified, it is not clear how those values can be translated into actions for development.
- There is no discussion based on the research findings. I would encourage the authors to include a discussion of their results in the paper.
Reviewer 2 Report
This manuscript reports the results of a study in Chiapas, Mexico, examining water services in relation to local needs, health outcomes, and other respects. The research is based on fieldwork with local Indigenous communities as well as key stakeholders involved in water treatment and services in the region. The results of the study are interesting and provide important insight into ways to re-frame discussions about water services, to get beyond divergent views of the significance of the problem and the root causes.
As written, however, that message is obscured by a great deal of jargon and generic statements. Furthermore, the study’s claim to be “transdisciplinary” is questionable. The study is multi-disciplinary, drawing on different academic theories, and the Indigenous communities were involved, but I did not see evidence that the communities themselves helped identify the research question or develop the methods or otherwise participate beyond the usual role of providing information. Terms like “transdisciplinary” will lose meaning if they are applied simply to claim good behavior by researchers, rather than being used only when a strict list of criteria is met. Please look carefully at the literature on “transdisciplinary.” If you still think this study meets that standard, then explain exactly how it does so. There is no problem if the study is not transdisciplinary, but only if it makes a claim to what it is not.
Major comments
First, ideas such as “value cocreation” and “service logic” (e.g., lines 110-112) are intriguing, but are not explained. One of the challenges of linking theories from different disciplines is that few readers will be familiar with all of these terms and ways of thinking. This means you will have to explain each term carefully. Readers should not have to look them up to understand your paper. Without an understanding of these terms, which appear to be new to the water treatment literature, readers may miss the point of your paper and may not realize all the implications of what you have found. That would be a shame, because it appears you have found something new and valuable.
Second, it seems to me that the main message of your paper is stated at lines 557-560. If this is correct, then it needs more prominence and attention. Sections 4.2.4 and 4.2.5, for example, say very little about this study, and mostly repeat common sentiments about the importance of working closely with local people, etc. See also lines 635-8: isn’t this what everyone says? The question is, what is stopping this from happening, or what is stopping it from working? Section 4.2.5 makes it sound like everyone is doing the right thing, so where’s the problem? If I understand this paper correctly, the problem is that everyone is trying to do the right thing, but using the wrong framework. See again lines 557-560, which point to a different way of looking at the problems and the solutions. This, it seems to me, is the message that needs to come through much more clearly. Then the rest of the Discussion can support this finding and point out what it would take to put it into practice. That is, what needs to change, from what is happening now, to what should happen with a new way of approaching the problem?
In sum, you have found something important and new to apply to the world of water. Please consider how you can re-organize and re-write the paper to make that finding come across clearly.
Minor comments
Lines 43-45: what about comparing to the same areas of Latin America and the Caribbean, rather than to the whole world?
91-93: this sentence is unclear
95-97: also not-for-profit organizations? If not, why not?
109: missing an “n” in Dominant (I did not mark all the typos and grammatical and punctuation errors—the manuscript needs a careful copy-editing)
113-115: incomplete sentence
Table 1: I don’t understand how to read this table. You refer to it early on, and then provide more information later, but I’m still confused. Columns 3-5 seem to align in rows, but Columns 1 and 2 do not fit this pattern. It is not clear what Column 2 adds, since the themes are repeated in Column 4. In Column 4, “Spiritual dimension” appears to be repeated in the section that goes with “Team Management” from Column 3. Is Column 3 in some kind of order, for example greater to lesser cocreation? Or what is the logic? Please think carefully about what this table is intended to portray.
171-173: re-word, for example changing “dependent” to “and depends” so it’s clear Chiapas is the poorest state overall, not just the poorest agriculture-dependent state (although if there are non-agricultural states that are poorer, you should re-word in a different way so it’s not ambiguous).
203: is “y” part of the group’s name, or should it be “and”?
253: “in lieu of”? As in, “in place of”? Or perhaps “as a counterpart to”?
263-6: what characteristics did you seek? The number of participants is pretty small (n=10 total), so it’s important to specify the characteristics of those who were in the IC focus groups. Ages? Roles in the community? Comparison to the community as a whole? The Results suggest that the focus group participants are older (see the disparaging comments about youth) and also see themselves somewhat different from the “average” resident (e.g., lines 370-3). So it’s important that the reader have more information about the IC participants.
Figure 4: saying that the problem is the trunk of the tree implies that there is only one problem. Do you use multiple trees if there is more than one problem? But of course the same effect can come from more than one problem, and more than one problem can come from the same cause. So I’m not sure how this tree is supposed to work. I’m also not sure what the figure adds to the paper, so it might be simplest just to delete it.
287-291: repeats what was said at 281-285. Please replace this with the correct text for Section 3.1.3.
297-298: wasn’t that the premise of the study? If not, what did you start with? Why did you do this study in the first place? It’s fine to confirm that your assumption was correct, but this seems odd to me as the key finding.
303: “lead author” isn’t a source. Perhaps “lead author’s observations” or “lead author’s other research on the topic”?
354: as far as I could tell, this is the first mention of the Maya. Please explain the relationship of the Maya to the Indigenous groups listed at line 203, for readers unfamiliar with the cultural landscape of Chiapas.
617: “created” not “crated”
635-638: isn’t this what everyone says? Again, as I pointed out in Major Comments above, please emphasize your findings, especially your valuable new insights.
661-663: did the participants say so? It’s easy for researchers to make claims like this, and sometimes the participants do say things like this. But I suspect the participants may often say nice things to visitors, so I’m distrustful of statements of this kind, in my own research, too. It sounds like the researchers patting themselves on the back. Best to avoid this. The work should speak for itself—you have good and interesting results because people were willing to talk openly with you. I have never seen a project where everything went smoothly and perfectly the whole time. Implying that it did creates a misleading impression for future researchers. If you think your methods are a new approach that is worth telling others about, I encourage you to write a separate “methods” paper in which you can explore those details, and tell the whole story, not just the “feel-good” story that emerges at the end of a project.
Round 2
Reviewer 1 Report
Thank you for the revisions. The paper is now in a better form. I would recommend it for acceptance.
Author Response
Response to Reviewer 2 Comments
Minor comments:
Point 1: "The only very minor error I noticed is in Table 1 (line 303) where Aged 36-45 years says n=1, but one male and one female are indicated below, and Aged 46-55 which says n=2 but only one male is indicated below. Please check which is correct and make the appropriate correction to the table."
Response 1: Many thanks for pointing out the mistake. The correct number for each age group is as follows:
Aged 36-45 (n=2)
Aged 46-55 (n=1)
The appropriate changes were made to the table inserted in the manuscript. Also, a corrected version of the table was added to the zip file containing the tables and figures accompanying the manuscript.
Reviewer 2 Report
I thank the authors for their extensive revisions and for considering my comments carefully. I am fully satisfied with the changes they have made, and I agree with their reasons for not following the comments they disagreed with. This paper is a valuable contribution and I look forward to being able to cite it in my own work.
The only very minor error I noticed is in Table 1 (line 303) where Aged 36-45 years says n=1, but one male and one female are indicated below, and Aged 46-55 which says n=2 but only one male is indicated below. Please check which is correct and make the appropriate correction to the table.
Well done and thank you for sharing your excellent study and insights!
Author Response
Response to Reviewer 2 Comments
Minor comments:
Point 1: "The only very minor error I noticed is in Table 1 (line 303) where Aged 36-45 years says n=1, but one male and one female are indicated below, and Aged 46-55 which says n=2 but only one male is indicated below. Please check which is correct and make the appropriate correction to the table."
Response 1: Many thanks for pointing out the mistake. The correct number for each age group is as follows:
Aged 36-45 (n=2)
Aged 46-55 (n=1)
The appropriate changes were made to the table inserted in the manuscript. Also, a corrected version of the table was added to the zip files containing the tables and figures accompanying the manuscript.